# Dual-Space Semantic Synergy Distillation for Continual Learning of Unlabeled Streams

**Donghao Sun**[*]    **Xi Wang**[*]    **Xu Yang**[†]    **Kun Wei**    **Cheng Deng**

School of Electronic Engineering, Xidian University, Xi'an 710071, China
{donghaosun508, wangxi6317, xuyang.xd, weikunsk, chdeng.xd}@gmail.com

## Abstract

Continual learning from unlabeled data streams while effectively combating catastrophic forgetting poses an intractable challenge. Traditional methods predominantly rely on visual clustering techniques to generate pseudo labels, which often suffer from semantic inconsistencies and limited discriminative precision, thereby impeding stable model evolution. To surmount these obstacles, we introduce an innovative approach that synergistically combines both visual and textual information to generate dual space hybrid pseudo labels for reliable model continual evolution. Specifically, by harnessing the capabilities of large multimodal models, we initially generate generalizable text descriptions for a few representative samples. These descriptions then undergo a 'Coarse to Fine' refinement process to capture the subtle nuances between different data points, significantly enhancing the semantic accuracy of the descriptions. Simultaneously, a novel cross-modal hybrid approach seamlessly integrates these fine-grained textual descriptions with visual features, thereby creating a more robust and reliable supervisory signal. Finally, such descriptions are employed to alleviate the catastrophic forgetting issue via a semantic alignment distillation, which capitalizes on the stability inherent in language knowledge to effectively prevent the model from forgetting previously learned information. Comprehensive experiments conducted on a variety of benchmarks demonstrate that our proposed method attains state-of-the-art performance, and ablation studies further substantiate the effectiveness and superiority of the proposed method.

## 1 Introduction

Deep learning models have demonstrated robust and well-established performance when trained on independently and identically distributed data, but real-world data is often nonstationary and arrives sequentially in tasks. In such scenarios, the model must continually learn new tasks while retaining knowledge of previous ones to avoid catastrophic forgetting [1, 2]. This learning paradigm is known as continual learning (CL), among which class-incremental learning (CIL) [3, 4] is particularly challenging and realistic, as it requires the model to perform unified classification while new classes are introduced progressively.

In many real-world scenarios, due to the high cost or even the infeasibility of labeling, data generally arrives in a continuous, unstructured, and unlabeled manner [5]. This raises a critical challenge: how can we perform structured modeling of continual streaming data in a fully unsupervised setting? Unlike conventional unsupervised learning [6], what makes the problem thornier is that unsupervised

---

[*]These authors contributed equally.
[†]Corresponding author.

39th Conference on Neural Information Processing Systems (NeurIPS 2025).

continual learning demands that models incrementally extract semantic structures over time, continually adapt to new data, and retain previously acquired knowledge. In this context, Unsupervised Class-Incremental Learning (UCIL) specifically targets the problem of progressively discovering class structures without any label supervision, and serves as a key step toward large-scale open-world learning.

In unsupervised learning, especially when dealing with large-scale unlabeled data, clustering-based pseudo-labeling is commonly used to guide model training [7]. However, existing clustering algorithms often perform poorly in complex visual scenarios, especially when dealing with visually similar categories (e.g., as illustrated in Fig.1a), where samples from different classes are easily confused. This frequently results in many incorrect pseudo-labels, severely hindering the model's ability to learn accurate feature representations. Such biased labels not only compromise training performance at the current stage but also cause irreversible knowledge corruption [8] in UCIL tasks, making it difficult for the model to correct early-stage misconceptions in later learning phases.

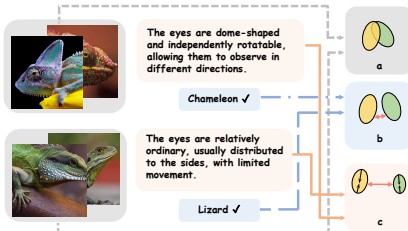

Figure 1: Classification results under different guidances. a) Visual-only (confusion risk) b) Coarse semantic (oracle labels) c) Fine-grained (optimized descriptions)

With the rapid advancement of large multimodal models (LMMs) [9], researchers have begun leveraging synthetic texts generated by LMMs as auxiliary supervision signals or contextual information to enhance overall performance [10–12]. Inspired by these developments, we explore the potential of using LMMs to uncover the latent semantic information within samples to further improve model effectiveness. We argue that natural language descriptions inherently contain semantic structures that can compensate for visual ambiguities (as illustrated in Fig.1b). Moreover, the relative stability of the semantic space provides a useful inductive bias for constraining model training, which in turn helps mitigate the problem of catastrophic forgetting [13].

Experimental results (Fig.2) demonstrate that directly employing LMMs as supervisory signals fails to enhance model performance while increasing computational complexity. Specifically, the standard unsupervised framework, fine-tuning through text clustering of LMM-generated image labels, shows limited performance gains. We attribute this to two factors: 1) The coarse-grained semantic representations from LMMs inadequately capture nuanced distinctions among visually similar samples [14]; 2) Inherent knowledge biases and inter-category semantic ambiguity [15] lead to semantic inconsistencies and inaccuracies in the generated pseudo labels, which critically constrain the efficacy of textual supervision.

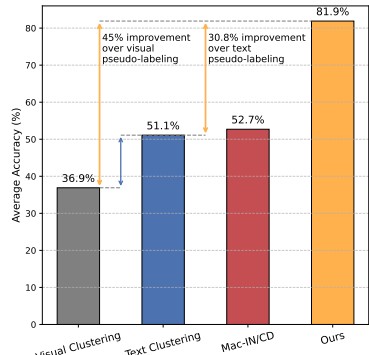

Figure 2: Quantitative experiments on ImageNet-R with 5 tasks.

*Inherent knowledge bias* refers to the tendency of large-scale pre-trained models (e.g., CLIP or large language models) to rely more heavily on concepts that appear frequently in their pre-training corpus during open-world reasoning. Consequently, these models often yield biased predictions for low-frequency or visually indistinct categories [16]. For example, frequent animals such as "cat" and "dog" are consistently recognized due to their salient visual features and stable semantic expressions, whereas rarer species like "weasel" or "lynx" tend to suffer representational drift because of limited occurrence or ambiguous semantics in the training data. This reflects the statistical bias inherent in the "language model as knowledge base" paradigm, where prior exposure determines the reliability of recognition [17].

*Inter-category semantic ambiguity* arises when categories share overlapping attributes in the visual or textual domain, blurring their semantic boundaries. For instance, "seal" and "sea lion" exhibit highly similar visual traits (body shape, texture, and background), making them difficult to separate visually, while "octopus" and "jellyfish"—though visually distinct—are often semantically conflated by descriptors such as "marine animal" or "tentacles" [18]. These within- and cross-modality ambiguities increase the likelihood of label confusion when LMM-generated descriptions are used directly for supervision.

In light of the above constraints, we select a small subset of representative samples (approximately 2% of the dataset) based on image clustering, and query LMMs to generate multi-granularity descriptions ranging from coarse to fine levels. This low-query strategy significantly reduces computational cost while enhancing the richness and discriminability of the extracted semantic information (as illustrated in Fig.1c). Meanwhile, considering the complementary strengths of visual and semantic modalities in perceptual granularity and representational capacity, we propose a language-guided collaborative alignment strategy to bridge and integrate supervisory signals from both spaces. By introducing language as an intermediary, we establish a connection between these two imperfect sources of pseudo-supervision, enabling the model to learn discriminative features that jointly capture abstract semantics and fine-grained visual details. Finally, as the semantic space excels at capturing abstract concepts and prior knowledge, while the visual space is more effective at modeling low-level details such as textures and edges, we introduce a semantic-to-visual distillation mechanism that leverages the stability of the semantic space to regularize model updates. This cross-modal alignment strategy effectively addresses the challenges of supervision scarcity and catastrophic forgetting in UCIL.

Our main contributions can be summarized as follows:

- We propose a low-resource UCIL approach that generates hierarchical semantic descriptions from LMMs using a small set of representative samples, effectively reducing computational cost and improving the reliability of pseudo labels, thereby enhancing the quality of semantic supervision.

- We design a dual-modality pseudo-labeling strategy that jointly leverages visual and semantic cues for robust representation learning, and introduce a semantic distillation mechanism to effectively mitigate catastrophic forgetting.

- Through comprehensive experiments and ablation studies, our method demonstrates superior performance and robustness on multiple benchmark datasets, highlighting its advantages in UCIL scenarios.

## 2 Related works

### 2.1 Unsupervised Class Incremental Learning

Unsupervised Class Incremental Learning (UCIL) focuses on learning new classes from unlabeled data while preserving knowledge of previously learned classes [19]. The challenge lies in preventing catastrophic forgetting as new classes are introduced. Self-supervised learning (SSL) is often employed [20], where models learn useful representations without labeled data. Contrastive learning, a popular SSL approach, distinguishes between positive and negative samples, enabling the model to learn discriminative features for new classes.

To address forgetting, techniques like memory replay and knowledge distillation are used [21]. Memory replay stores previously encountered data and replays it during training to maintain performance on old classes while learning new ones [22]. Additionally, combining unsupervised learning with pseudo labels or cross-modal information can help improve learning efficiency and task adaptability in UCIL settings [23].

### 2.2 Fine-Tuning CLIP Models

CLIP models, pretrained on large-scale image-text pairs, have demonstrated strong zero-shot performance, but fine-tuning is often necessary to improve task-specific performance [24]. A common approach to fine-tuning CLIP involves training a classifier on top of the pre-trained features. One efficient method is adding a fully connected linear layer at the output of CLIP's visual encoder. This approach freezes the CLIP model's parameters and only fine-tunes the linear layer, minimizing the risk of overfitting and reducing the computational cost compared to full fine-tuning [25]. This method is effective in adapting CLIP to specific tasks while maintaining the robustness of its pre-trained representations.

Recently, methods such as contrastive loss [26] and linear probing [27] have also been applied to CLIP models for task adaptation. However, freezing the core CLIP model while fine-tuning a small number of parameters, like the output layer, has emerged as a popular technique, as it balances computational efficiency with high performance in specific applications.

# 3 Method

The objective of this work is to enable the network to learn from continuous, unlabeled data streams that more accurately reflect the real-world environment. The proposed method efficiently utilizes the cross-modal information inherent in the data streams, while minimizing resource consumption. It is anticipated that this approach to cross-modal information alignment will offer a more comprehensive solution for future unsupervised tasks. In this section, we first present the definition of UCIL, followed by a detailed explanation of the method we have introduced.

## 3.1 Preliminary

### 3.1.1 Problem definition.

In the context of UCIL, the model needs to be trained on $T$ consecutive tasks. Each task $t$ provides an unlabeled dataset $D^t = \{x_i\}_{i=1}^{N^t}$, where $N^t$ represents the number of instances in task $t$, and these instances belong to $C^t$ new categories. The categories across tasks are disjoint, meaning there is no overlapping between any two different tasks: $C^{t_i} \cap C^{t_j} = \varnothing, i \neq j$. The goal of UCIL is to enable the model to incrementally discover semantically meaningful categories from $D^t$ and assign instances to these categories without label information. In each task, the model has access only to the current unlabeled dataset $D^t$ and cannot access any prior information. The number of new categories $C^t$ introduced in task $t$ is known in advance, but for the specific category instances of each task, the model can only discover them through the exploration of the data. During the incremental learning process, the model not only needs to learn the new categories in the current task but also must retain the categories learned previously, preventing the forgetting of knowledge from earlier tasks. To formalize this problem, let $X$ be the input data space. The objective is to learn a mapping function $f : X \to \bigcup_{t=1}^{T} C^t$, which can map any test sample $x$ to the set of categories discovered across all tasks without relying on task identifiers.

### 3.1.2 Base model

CLIP was chosen as the base pre-trained model for our work due to its ability to simultaneously process data from both visual and linguistic spaces. An effective approach to fine-tuning a pre-trained model for downstream tasks involves incorporating a lightweight network as an adapter. To facilitate our subsequent explanation, we denote the visual encoder as $E_v$, the text encoder as $E_t$, and the linear adapter as $f$. Under supervised conditions, when textual labels are employed as supervision information for category classification, the probability that the model predicts any input sample $x_i$ as category $l_k$ is:

$$\texttt{pred}_k = E_t(l_k) \cdot f(E_v(x_i)). \tag{1}$$

In the previous analysis, we performed classification in the visual and textual knowledge spaces using traditional clustering methods, but the results were unsatisfactory. Even though we obtained information from both modalities in the data stream, the high visual similarity between categories and the inherent semantic ambiguity in the textual information derived from LMMs made it difficult to achieve satisfactory classification performance with existing methods. Therefore, in this paper, we propose two approaches to address the confusion within tasks caused by the lack of supervisory signals, as well as the confusion between different tasks induced by incremental data. The architectures of these methods are illustrated in the Fig.3. We will now provide a detailed description of the proposed methods.

## 3.2 Semantic Collaborative Facilitative Supervision

To fully exploit cross-modal information, we avoid performing clustering independently in a single modality. Instead, we propose semantic collaborative supervision, which aligns visual and semantic knowledge in a unified framework. Specifically, we reduce the reliance on LMM queries by selecting only a few representative samples near visual cluster centers, rather than querying every instance. To enhance the quality of semantic signals, we design a series of progressively refined prompt templates for hierarchical description extraction. The resulting visual and textual cues are then jointly used in a collaborative supervision mechanism to guide model training more effectively.

For task $t$, we process the input samples $x_i$ through the CLIP visual encoder to obtain their feature representations $E_v(x_i)$ in the visual space. We then apply K-means clustering in this space to partition

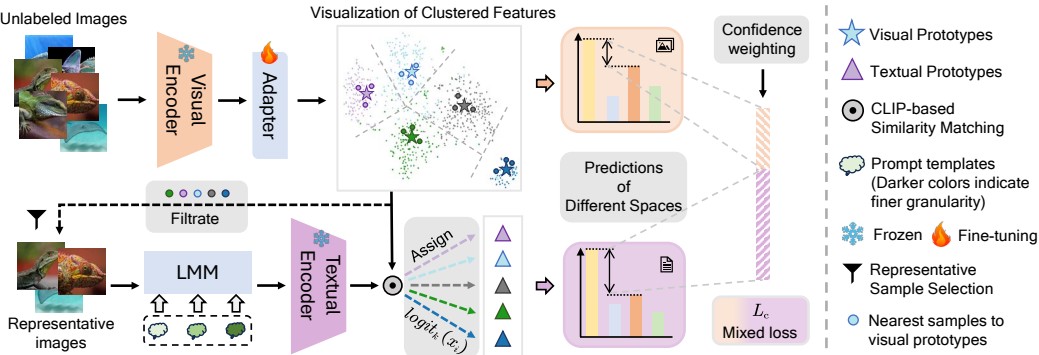

Figure 3: Framework for a single task in unsupervised class-incremental learning. Visual clusters and LMM-generated descriptions provide dual-modality pseudo-labels, which jointly supervise adapter training via confidence-weighted loss.

the samples into $C^t$ clusters, denoted as $\{\phi_k\}_{k=1}^{C^t}$. For the $k$-th cluster $\phi_k$, we treat the samples that are closest to the cluster center as belonging to the $k$-th class and assign them the corresponding visual pseudo-label $p_v$. Since the visual feature space may contain similar but semantically different image representations, the resulting visual pseudo labels often suffer from inter-class confusion. Using such confusing pseudo labels as supervision to fine-tune the pre-trained model may lead to an unstable optimization process, hindering transferability to downstream tasks. Therefore, it is necessary to introduce more reliable supervisory signals. Given the superior stability and generalizability of the semantic space, we leverage independently designed textual descriptions to filter the data and reduce the class confusion in the visual pseudo labels.

To obtain text descriptions, the most straightforward method is to directly input these samples into an LMM and prompt it with the query, *'What object is depicted in this image?'*, which generates a coarse-grained category description for each sample $x_i$. Besides the additional time and financial costs associated with utilizing the LMM, the generated text may still exhibit a degree of randomness due to the LMM's inherent knowledge biases and the potential ambiguity between different classes. Under such textual supervision, this randomness may exacerbate the confusion between samples. Therefore, based on the previous clustering results, for any given cluster $k$, we first obtain the prototype corresponding to each cluster $k$:

$$\mu_k = \frac{1}{n_k} \sum_{i=1}^{n_k} E_v(x_i), x_i \in \phi_k, \tag{2}$$

where $n_k$ represents the number of samples in $\phi_k$. Then, we calculate the similarity between all samples in the cluster and the cluster prototype to construct the similarity matrix $Q_k = [q_i]_{i \in n_k}$ for any $\phi_k$, $q_i$ represent the cosine similarity score corresponding to each sample,

$$q_i = \cos\left(E_v(x_i), \mu_k\right). \tag{3}$$

Based on the similarity matrix, we can obtain the most representative $m$ samples from each clustering result. We consider these $m$ samples to have high-quality representations of each cluster. By providing fine-grained text descriptions for these samples, we avoid semantic ambiguity introduced by outlier or boundary samples and reduce computational resource consumption.

At the same time, to reduce the randomness caused by sample selection and obtain more generalizable text descriptions, we designed the following three prompt templates in a sequence from coarse-grained to fine-grained to guide the LMM in generating the corresponding text descriptions.

**Prompt 1:** *'Please tell me the name of the object in the image without any descriptors.'* This prompt quickly captures the main object in the image by directly asking for its basic name, avoiding redundant information.

**Prompt 2:** *'Please describe the most distinctive visual attributes in the photo.'* This prompt focuses on the fine-grained visual features of the image, helping to identify subtle details or distinctive visual markers that may not be immediately noticeable.

**Prompt 3:** *'Please describe the most common scenes of the object in the photo.'* This prompt describes the common scenes of the object, providing a coarse-grained background, usage, and context.

Under the guidance of these three prompts, the LMM generates a total of $\{l_i^k\}_{i=1}^{3m}$ textual descriptions for the $m$ samples in each clustering cluster $\phi_k$. These texts are treated as the textual descriptors for independent classes. Based on these textual descriptors, we can perform fine-grained filtering of the visual features. For any given input sample $x_i$, the probability of it belonging to the $k$-th class is given by:

$$\texttt{logit}_k(x_i) = \frac{1}{3m} \sum_{j=1}^{3m} E_t(l_j^k) \cdot E_v(x_i). \tag{4}$$

We consider the class with the highest probability as the textual pseudo-label $p_t$ for the input sample. By using fine-grained textual descriptions, we optimize the initial purely visual labels and leverage the stability of the semantic space to provide higher-quality supervision signals for unsupervised tasks. For any given sample, we assign its corresponding visual pseudo-label $p_v$ and textual pseudo-label $p_t$. With the assistance of these supervision signals, we can thereby transform the originally unsupervised task into a supervised one and optimize the model using the cross-entropy loss $\mathcal{L}_{ce}(x_i, p)$.

Textual pseudo labels, derived from the semantic space, provide high-level and stable representations but may overlook fine-grained visual differences. When generated by LMMs, they can also exhibit semantic bias or inaccuracy. In contrast, visual pseudo labels capture detailed features but are more sensitive to inter-class confusion and ambiguity. To leverage their complementary strengths and offset their limitations, we propose a hybrid supervision strategy that combines both modalities. Inspired by the general principle of multimodal fusion that higher-confidence predictions tend to provide more reliable supervisory signals [28], our visual–semantic dynamic weighting mechanism adaptively balances the two modalities according to their relative confidence. Increasing the weight of the more confident modality helps stabilize training and improve effectiveness. Based on this intuition, we design a sample-level adaptive weighting strategy that fuses supervision from the visual and semantic spaces according to their confidence difference. For each sample $x_i$, the training loss is defined as follows:

$$\mathcal{L}_{\text{cls}}^i = \omega_v^i \mathcal{L}_{\text{ce}}(x_i, p_v) + \omega_t^i \mathcal{L}_{\text{ce}}(x_i, p_t). \tag{5}$$

For the input sample $x_i$, if the clustering results in the visual space and textual space are identical, meaning that $p_v = p_t$, we consider the sample to be in a stable state. In this case, we consider the additional information provided by both the visual and textual spaces to be of equal importance.

For cases where the clustering results differ between the two modalities, $p_v \neq p_t$, we consider the sample to be in a state of confusion. In this case, we need to find the balance between the visual and textual spaces. To achieve this, we first calculate the difference between the maximum probability and the second maximum probability at the stage of assigning pseudo labels to samples.

In the visual space, for any sample $i$, we first compute the cosine similarity between its feature and all class prototypes in the current task:

$$s_j^i = \frac{E_v(x_i) \cdot \mu_j}{\|E_v(x_i)\| \, \|\mu_j\|}, \quad j = 1, \ldots, C^t. \tag{6}$$

The confidence margin for sample $i$ in the visual space is then defined as the difference between the top-1 and top-2 similarities:

$$\mathcal{H}_v^i = \texttt{top}_1(s_j^i) - \texttt{top}_2(s_j^i). \tag{7}$$

Similarly, in the textual space, the confidence margin is computed as:

$$\mathcal{H}_t^i = \texttt{top}_1(\texttt{logit}_j(x_i)) - \texttt{top}_2(\texttt{logit}_j(x_i)), \quad j = 1, \ldots, C^t. \tag{8}$$

This difference, after normalization, is regarded as the reliability of the supervisory information provided by the pseudo labels from different spaces:

$$\omega_{\{v,t\}} = \frac{\mathcal{H}_{\{v,t\}}^i}{\mathcal{H}_v^i + \mathcal{H}_t^i}. \tag{9}$$

Therefore, under the supervision of visual-text dual pseudo labels, the model's training loss is given by:

$$\mathcal{L}_c^i = \begin{cases} \frac{1}{2}\mathcal{L}_{ce}(x_i, p_v) + \frac{1}{2}\mathcal{L}_{ce}(x_i, p_t) & \text{if } p_v = p_t \\ \frac{\mathcal{H}_v^i}{\mathcal{H}_v^i + \mathcal{H}_t^i}\mathcal{L}_{ce}(x_i, p_v) + \frac{\mathcal{H}_t^i}{\mathcal{H}_v^i + \mathcal{H}_t^i}\mathcal{L}_{ce}(x_i, p_t) & \text{else.} \end{cases} \quad (10)$$

Finally, we perform a weighted summation for a batch of training samples $\mathcal{L}_{cls} = \sum_i \mathcal{L}_c^i$, which serves as the loss of semantic supervision.

By balancing the reliability of supervisory information from the visual and textual spaces, we effectively mitigate the inconsistency between the two supervisory signals, maximizing the utility of pseudo labels. This, in turn, enhances the success rate of transferring the model to downstream tasks. The effectiveness of this weighting mechanism is further validated by our experiments in Table 3.

### 3.3 Semantic Alignment Distillation

Due to the influence of continuous data streams, the constantly changing visual features affect the feature space constructed by previous data, leading to catastrophic forgetting in the model. Compared with the instability of visual representations, the semantic space offers stronger structural consistency and scalability. To leverage this advantage, we propose Semantic Alignment Distillation (SAD), which utilizes the stability of the language space to regularize the evolution of the visual space and mitigate catastrophic forgetting.

After each task training, we store the class prototypes $\mu_k = \frac{1}{n_k}\sum_{i=1}^{n_k} E_v(x_i)$ and covariance matrices $\Sigma_k$ for all categories based on the textual space prediction results. During subsequent tasks, for each training batch, we randomly sample $c$ instances from the Gaussian distribution of old data. For these sampled instances, we compute the similarity $\texttt{logit}_k(x_i^{old})$ between each sample and its corresponding textual descriptor, as well as $\texttt{logit}_g(x_i^{old})$ between the same sample and the textual descriptor of a randomly selected new class. Here, $k$ and $g$ denote the categories of the old and new classes, respectively. Given the invariance of textual representations, we define the semantic alignment loss for each sample as:

$$\mathcal{L}_s^i = 1 - \texttt{logit}_k(x_i^{old}) + \texttt{logit}_g(x_i^{old}). \quad (11)$$

The first term in Eq. 11 measures the similarity between the sample and its corresponding textual prototype, while the second term evaluates its similarity to a randomly selected new class prototype. This formulation encourages old samples to remain close to their semantic anchors while keeping a margin from new class descriptors, thereby constraining the visual representation space during incremental updates. The semantic alignment loss for a batch is expressed as $\mathcal{L}_{sal} = \sum_{i=1}^{c} \mathcal{L}_s^i$.

To further stabilize feature evolution, we add a distillation constraint between the previous and current adapters:

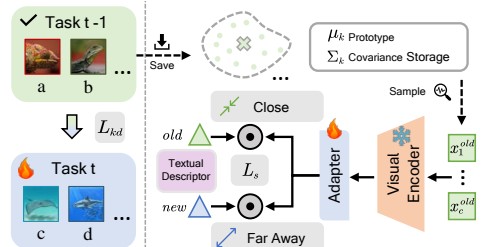

Figure 4: Semantic distillation leverages stable textual descriptors to constrain visual training and mitigate catastrophic forgetting.

$$\mathcal{L}_{kd} = \|f_{old}(E_v(x)) - f_{new}(E_v(x))\|_2. \quad (12)$$

The overall training objective integrates three components:

$$\mathcal{L} = \mathcal{L}_{cls} + \lambda_1 \mathcal{L}_{sal} + \lambda_2 \mathcal{L}_{kd}. \quad (13)$$

Different from prototype-based distillation methods that operate solely within the visual space, SAD introduces a cross-modal alignment mechanism that transfers stability from the semantic space to the visual domain. It constructs a language-driven semantic scaffold composed of textual prototypes generated via hierarchical prompts (Sec. 3.2), whose embeddings are obtained from a frozen text encoder and remain invariant throughout learning. These prototypes act as semantic anchors that define a stable manifold for guiding visual feature updates. By enforcing directional alignment between visual features and their fixed textual counterparts, SAD emphasizes semantic consistency rather than visual reconstruction, providing a novel and effective approach to mitigating catastrophic forgetting in unsupervised class-incremental learning.

# 4 Experiments

## 4.1 Experiments Settings

**Datasets.** We conducted experiments on three datasets, including the widely used image classification dataset CIFAR100, the style-varied classification dataset ImageNet-R, and the fine-grained dataset CUB200. The CIFAR100 dataset contains 100 distinct classes, while both the ImageNet-R and CUB200 datasets consist of 200 different classes. We partitioned each of the three datasets into 5, 10, and 20 consecutive task streams, with an equal distribution across the tasks.

**Metrics.** In continual learning, performance is commonly evaluated using two metrics: the *Last Stage Accuracy (LA)* and the *Overall Average Accuracy (AA)*, following common practice in class-incremental learning [3]. The average accuracy is computed as $A = \frac{1}{T} \sum_{t=1}^{T} A_t$, where $A_t$ denotes the accuracy on all seen classes after learning task $t$, and $T$ is the total number of tasks. The final accuracy (LA) is defined as $A_T$, which measures the performance on all classes after completing the last task, reflecting the model's overall retention capability. In the main paper, we report the final accuracy (LA) of each method for clarity and fairness of comparison. To provide a more comprehensive evaluation, the corresponding AA results of our method are included in Appendix A.4.

**Comparison methods.** We compare our method with state-of-the-art algorithms, including unsupervised class incremental learning methods CaSSLe [20], PFR [29], POCON [30] and MSc-iNCD [31]. Unsupervised representation learning UPS [32]. The original term MSc-iNCD refers to novel class discovery, but its setup is identical to the UCIL in this paper. For unsupervised representation learning methods, we directly trained the model using the UPS algorithm under two scenarios: one without any constraints and the other with adding knowledge distillation. Meanwhile, we directly use CLIP to cluster the test data, and we also labeled all the samples in the test set using LMM and then clustered the text to enhance the comprehensiveness of our experiments [33].

**Implementation details.** For both datasets, our pretrained model is the ViT-L/14 version of CLIP, and we train the model with the Adam optimizer for 30 epochs, with a learning rate of $1 \times 10^{-3}$. We use CosineSchedule to adjust the learning rate. To ensure experimental fairness, all comparison methods were conducted in the same environment. In our experiments $m = 3$ and $\lambda_1 = 1$ for all datasets and $\lambda_2 = 0.03$ for ImageNet-R and CIFAR100, $\lambda_2 = 0.15$ for CUB200. For all datasets, the LMM that we used is GPT-4-turbo. In order to ensure the fairness of our experiment, we try our best to keep the training environment the same, and all the methods are obtained by re-running them on Python 3.8, Pytorch 2.0.1, and a single GPU A6000. All baselines were also re-run using the same ViT-L/14 backbone for fair comparison.

Table 1: Comparison experiments on different benchmarks. **Bold** indicates the best, while underline represents the second-best. Results are averaged over 20 random seeds for robustness verification (see Appendix A.4).

| Method | ImageNet-R | | | CIFAR100 | | | CUB200 | | |
|---|---|---|---|---|---|---|---|---|---|
| | 5 tasks | 10 tasks | 20 tasks | 5 tasks | 10 tasks | 20 tasks | 5 tasks | 10 tasks | 20 tasks |
| UPS [ICLR'21] | 20.9 | 16.5 | 14.3 | 18.4 | 15.7 | 12.9 | 18.3 | 15.9 | 13.1 |
| UPS + KD [ICLR'21] | 37.9 | 35.2 | 32.6 | 43.9 | 37.8 | 35.6 | 29.7 | 25.3 | 22.1 |
| CassLe [CVPR'22] | 45.2 | 40.5 | 37.8 | 59.6 | 52.5 | 49.6 | - | - | - |
| PFR [CVPR'22] | 41.6 | 37.9 | 31.0 | 59.8 | 54.3 | 44.8 | - | - | - |
| POCON [WACV'24] | 40.3 | 41.5 | 41.1 | 63.1 | 60.5 | 56.8 | - | - | - |
| MSc-iNCD [ICPR'24] | 52.7 | 53.4 | 51.2 | 64.9 | 62.7 | **60.3** | 32.9 | 31.9 | 30.4 |
| CLIP-based clustering | 36.9 | 36.9 | 36.9 | 32.9 | 32.9 | 32.9 | 31.6 | 31.6 | 31.6 |
| LMM-based text clustering | 51.1 | 51.1 | 51.1 | 41.7 | 41.7 | 41.7 | 34.5 | 34.5 | 34.5 |
| **Ours** | **81.7** | **82.2** | **79.8** | **66.1** | **63.6** | 59.4 | **66.7** | **64.8** | **64.0** |

## 4.2 Experimental Results

We conducted experiments on various datasets under different settings, and the results are shown in Table 1. On ImageNet-R with 5 tasks, our method achieves $81.7\%$, yielding a $29.0\%$ improvement over MSc-iNCD. This gain is not simply due to the use of CLIP, as the clustering result obtained by CLIP alone is only $36.9\%$. Similarly, clustering with LMM-generated text performs better than visual clustering but still falls short of the previous SOTA. When the number of tasks increases to 10

and 20, our method remains robust, achieving accuracies of $82.2\%$ and $79.8\%$. On the fine-grained CUB200 dataset, where distinguishing similar image features is particularly challenging, all prior methods show relatively low performance. In contrast, our method reaches $66.7\%$ on the 5-task setting, surpassing the previous SOTA by $33.8\%$, and maintains $64.8\%$ and $64.0\%$ with 10 and 20 tasks. For CIFAR100, our method achieves $66.1\%$, $63.6\%$, and $59.4\%$ with 5, 10, and 20 tasks, respectively. The relatively lower performance can be partly attributed to the $32 \times 32$ resolution of CIFAR100 images, which limits the LMM's ability to extract fine-grained semantics and generate accurate textual descriptions, thus affecting pseudo-label quality. In comparison, on higher-resolution datasets, our method consistently demonstrates strong and stable performance.

## 4.3 Ablation Study

In this section, we analyze the effectiveness of each component in our proposed method. The experiments were conducted on ImageNet-R across 5 tasks, with the results presented in Table 2. As shown in the table, although pre-trained models possess powerful representational capabilities, relying solely on clustering fails to achieve highly accurate classification results when faced with unsupervised data, clustering method based solely on visual features achieved an accuracy of $36.9\%$. This result does not meet the performance requirements. By treating the clustering results as a visual pseudo-label and fine-tuning the model under the constraint of distillation loss, the model ultimately achieved an accuracy of $50.4\%$. Due to the poor clustering performance in the previous step, the visual pseudo labels contain substantial inter-class confusion and labeling inaccuracies. As a result, relying solely on the supervisory information introduced by the visual space is insufficient to successfully transfer the pre-trained model to downstream tasks.

Considering the remarkable stability and knowledge expansion capability of textual information, we used our generated generalized text descriptors to filter the samples. Under this LLM-based textual supervision, where diverse and unstructured labels are generated for each image and clustered into category prototypes, the fine-tuned model achieved a performance of 76.8% on downstream tasks. Building upon this, our proposed hierarchical prompting strategy queries only a few representative samples per class to construct consistent and discriminative textual prototypes, and further introduces a SAD loss to enhance cross-modal consistency. This combined strategy improves the performance to 81.9%. For reference, using ground-truth labels to construct text prompts in the form of "a photo of a [label]" yields a supervised upper bound of 84.2%. This comparison demonstrates that our semantically guided pseudo-label design, even without any real labels, achieves performance close to the supervised upper limit.

Table 2: Ablation study on ImageNet-R, 5 tasks.

| | Components | | | | Accuracy |
|---|---|---|---|---|---|
| | $p_v$ | $p_t$ | $\mathcal{L}_{kd}$ | $\mathcal{L}_{lkd}$ | |
| a) | | | | | 36.9 |
| b) | $\checkmark$ | | $\checkmark$ | | 50.4 |
| c) | | $\checkmark$ | $\checkmark$ | | 76.8 |
| d) | $\checkmark$ | $\checkmark$ | $\checkmark$ | | 78.1 |
| e) | $\checkmark$ | $\checkmark$ | $\checkmark$ | $\checkmark$ | 81.9 |

To maximize the utilization of supervisory information from both modalities, we further reduce pseudo-label inaccuracies by adaptively weighting each space according to its reliability. On the ImageNet-R benchmark, as shown in Table 3, this balanced dynamic weighting strategy outperforms the fixed weighting scheme, providing strong evidence for the effectiveness of the proposed method.

Table 3: Dynamic vs. fixed weighting.

| Tasks | Dynamic Weighting (%) | Fixed Weighting (%) |
|---|---|---|
| 5 | 81.7 | 80.2 |
| 10 | 82.2 | 80.3 |
| 20 | 79.8 | 77.1 |

## 4.4 Further Analysis

**Annotating images with LMM.**

We analyzed the initially proposed approach of directly employing a LMM to annotate the entire dataset and conducted experiments on the ImageNet-R benchmark. The results indicate that LMMs are highly susceptible to interference from complex backgrounds and environmental variations, often leading to inaccurate predictions(as exemplified by the misannotations shown in Fig. 5). This

naive annotation strategy not only lacks reliability but also incurs substantial computational costs. These findings underscore the necessity and effectiveness of the proposed Semantic Collaborative Facilitative Supervision framework introduced in this work.

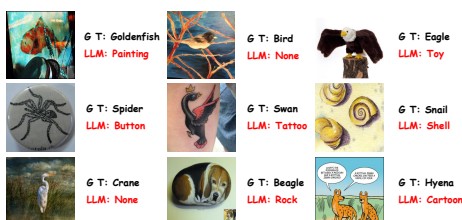

Figure 5: Incorrect labels generated by the LMM.

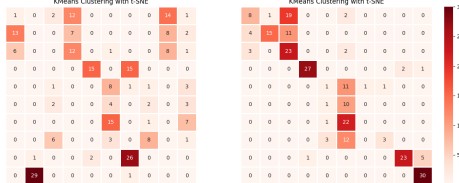

Figure 6: The confusion matrix of pseudo labels: the left represents visual pseudo labels, while the right represents textual pseudo labels.

**Comparison of pseudo-label prediction results.**

In this paper, we propose a dual pseudo-labeling strategy for both visual and textual spaces. We randomly selected 10 classes from ImageNet-R and constructed the confusion matrix shown in Figure 6, where the horizontal and vertical axes represent the true labels and pseudo-labels, respectively. Larger values along the diagonal indicate higher accuracy of the pseudo-labels. Experimental results demonstrate that textual descriptions, due to their stronger generalization capability, outperform visual clustering. However, both types of pseudo-labels still contain errors, which led us to design a balanced strategy that integrates predictions from both spaces to more effectively leverage supervisory information.

To further validate the robustness and adaptability of the proposed method, we include additional experimental analyses in the supplementary material, including, but not limited to: (1) the impact of the number of representative samples $m$ on the quality of LMM-generated descriptions; (2) the effect of different prompt; (3) sensitivity analysis of hyperparameters (e.g., $\lambda_1$ and $\lambda_2$). We also include additional analyses on the computational and resource overhead introduced by integrating the LMM component, comparing API-based and fully local deployment setups. Detailed results and cost breakdowns are provided in Appendix A.5.

## 5 Conclusion

In this paper, we propose a hierarchical, language-guided approach to generate fine-grained text pseudo labels. By combining these with traditional visual clustering pseudo labels, we create multimodal weighted text-visual pseudo labels to guide training, addressing the challenge of missing supervision in unsupervised class-incremental learning. We also introduce a collaborative alignment method that uses text pseudo-labels to constrain training and reduce catastrophic forgetting between incremental tasks. Our method is compared with various approaches in unsupervised class-incremental learning, and extensive experiments on benchmark datasets demonstrate its effectiveness. Additionally, we conduct ablation studies to validate the rationale behind our approach.

We sincerely hope that the proposed method offers a novel and impactful perspective for advancing the fields of unsupervised and incremental learning. By strategically harnessing the capabilities of modern multimodal models, our approach aims to unlock the latent structure within vast, unannotated real-world data streams. We believe this work serves as a step toward more intelligent, scalable, and label-efficient learning systems capable of adapting to the complexity and openness of dynamic environments.

## 6 Acknowledgment

This work is supported in part by the National Key Research and Development Program of China (No. 2023YFC3305600), Joint Fund of Ministry of Education of China (8091B022149, 8091B02072404), National Natural Science Foundation of China (62132016, 62571412, and 62571393), Key Research and Development Program of Shaanxi (2024GX-YBXM-127) and National Key Laboratory Foundation of China (Grant No. HTKJ2024KL504011).

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

# A    Appendix / supplemental material

To further assess the robustness, generalizability, and practical effectiveness of the proposed method, we present a series of additional experimental analyses in this supplementary material. These analyses offer deeper insights into the behavior of the model under varying conditions and validate the design choices made in the main paper. Specifically, we investigate the influence of key components and parameters that affect the performance of our approach.

## A.1    Impact of the Number of Representative Samples ($m$)

To assess how the number of representative samples $m$ affects the quality of LMM-generated descriptions and overall model performance, we conducted experiments on the ImageNet-R dataset with 5 tasks. The results are summarized in Table 4.

Table 4: Effect of the number of representative samples ($m$) on classification accuracy (ImageNet-R, 5 tasks).

| $m = 1$ | $m = 2$ | $m = 3$ | $m = 4$ | $m = 5$ |
|---------|---------|---------|---------|---------|
| 81.0 | 81.4 | **81.9** | 82.0 | 82.2 |

As shown in Table 4, increasing the number of representative samples generally improves performance. The accuracy increases steadily from 81.0% (when $m = 1$) to 82.2% (when $m = 5$). This improvement is attributed to the richer and more diverse descriptions generated by the LMM when provided with more query samples.

In our final setting, we choose $m = 3$ as a trade-off between accuracy and computational efficiency. Generating high-quality descriptions via LMM is relatively time-consuming, and larger values of $m$ incur additional query and processing overhead. The setting $m = 3$ achieves competitive accuracy (81.9%) while keeping the LMM query cost low, making it a practical and scalable choice for continual learning scenarios.

## A.2    Effect of Different Prompt Designs

To evaluate the impact of prompt design on the effectiveness of LMM-generated textual descriptions, we conducted a series of ablation experiments using multiple prompt templates. The goal is to determine the sensitivity of our method to prompt formulations and to explore whether certain prompt combinations lead to more informative and discriminative descriptions, ultimately improving classification performance.

We abbreviated the prompt templates as P, where P1, P2, and P3 were all derived from the main text. Additionally, we introduced P4 to increase the diversity of the experiment. P4 consists of the generic prompt "a photo of []", which was completed by directly querying the LMM, serving as a widely-used baseline template. The results of this experiment on the ImageNet-R dataset with 5 tasks are summarized in Table 5.

Table 5: Accuracy (%) of different prompt combinations on ImageNet-R (5 tasks).

| Prompt Combination | Accuracy (%) |
|--------------------|--------------|
| P1 + P2 + P3 | **81.9** |
| P1 | 79.4 |
| P2 | 77.6 |
| P3 | 76.9 |
| P1 + P2 | 80.8 |
| P1 + P3 | 80.1 |
| P2 + P3 | 78.4 |
| P4 | 79.0 |

These results demonstrate several key findings:

- **Prompt diversity helps**: Combining multiple prompts (P1+P2+P3) yields the best accuracy, suggesting that richer semantic descriptions enhance the model's ability to generalize.
- **Even simple prompts work**: Although P4 uses a very simple and generic format, it still achieves competitive performance (79.0%), outperforming many prior SOTA baselines, which highlights the robustness of our method to prompt variations.

Overall, this experiment confirms that while our method is relatively robust to different prompt formats, careful prompt engineering can further boost performance. Additionally, it illustrates that even basic prompts can produce effective representations when combined with our proposed learning framework.

### A.3 Sensitivity Analysis of Key Hyper-parameters ($\lambda_1$, $\lambda_2$)

In this section, we analyze how variations in the key hyper-parameters $\lambda_1$ and $\lambda_2$ affect the performance of our proposed method. These hyper-parameters play critical roles in balancing the contributions of visual and semantic modalities during training.

**Hybrid Loss Balancing Mechanism**

Our method employs a hybrid loss function that integrates visual and textual supervision. The balancing of this hybrid loss is adaptive and depends on the consistency between the predictions from the visual and semantic spaces. Specifically:

- If the predicted class labels from both modalities agree, we assign equal weights to both terms.
- If the predictions differ, the weights are adjusted dynamically based on the difference in confidence (i.e., probability gap between the top two predicted classes) in each modality.

This mechanism helps to assign higher weight to the more confident modality, thereby improving stability and reducing the impact of unreliable pseudo labels.

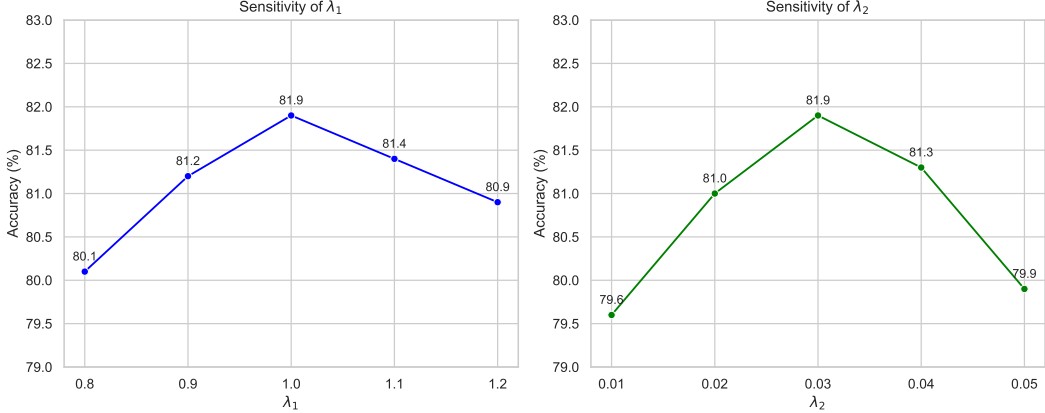

Figure 7: Sensitivity analysis of hyper-parameters $\lambda_1$ and $\lambda_2$ with respect to classification accuracy (ImageNet-R 5 tasks). Each bar represents the model's accuracy under a different value of the corresponding hyper-parameter.

**Sensitivity Analysis of $\lambda_1$**

We conducted a sensitivity analysis on the hyper-parameter $\lambda_1$, which governs the weight of the visual modality in the hybrid loss. The default setting was $\lambda_1 = 1.0$. To evaluate the robustness of this parameter, we varied its value within a small range, specifically testing values from 0.8 to 1.2 in increments of 0.1.

As shown in Figure 7, model performance remains relatively stable across this range. The accuracy peaks at the default value of $\lambda_1 = 1.0$, and only minor performance degradation is observed when

moving away from this setting. These results suggest that the method is robust to small perturbations in $\lambda_1$.

**Sensitivity Analysis of $\lambda_2$**

We further analyzed the sensitivity of $\lambda_2$, which controls the influence of the semantic distillation loss. The default value of $\lambda_2$ was set to 0.03. We explored its effect on classification accuracy by varying it between 0.01 and 0.05 in increments of 0.01.

As depicted in Figure 7, the model achieves the best performance at the default value. Although performance decreases slightly when $\lambda_2$ is adjusted by ±0.01, the drop remains within 1.3 percentage points, indicating relative stability in this parameter as well. However, $\lambda_2$ appears slightly more sensitive than $\lambda_1$, with a more pronounced performance peak at its tuned value.

**Discussion**

Overall, the results illustrated in Figure 7 demonstrate that the proposed method exhibits good robustness to moderate changes in both $\lambda_1$ and $\lambda_2$. Although the model is not highly sensitive to these hyper-parameters, careful tuning can still yield incremental improvements in classification performance.

## A.4 Multiple-Seed Evaluation and Accuracy Metrics

To further assess the stability and robustness of our proposed method, we conducted 20 independent runs with different random seeds on all three datasets, following the evaluation protocol described in Section 4.1. We report both the *Last Task Accuracy (LA)* and the *Average Accuracy (AA)*—two standard metrics widely used in class-incremental learning [3]. The results are summarized in Table 6, where each score represents the mean accuracy (%) ± standard deviation (%) over 20 runs. The small variances demonstrate the strong consistency of our approach under different initialization conditions, while the close gap between LA and AA indicates stable performance across incremental stages.

Table 6: Performance of our method under 20 independent runs, reporting both Last Task Accuracy (LA) and Average Accuracy (AA).

| Dataset | Tasks | LA (%) | AA (%) |
|---------|-------|--------|--------|
| ImageNet-R | 5 | 81.7±0.9 | 89.8±0.8 |
|  | 10 | 82.2±0.7 | 85.9±0.6 |
|  | 20 | 79.8±0.6 | 83.4±0.5 |
| CIFAR100 | 5 | 66.3±1.0 | 74.8±0.9 |
|  | 10 | 63.8±0.8 | 70.5±0.7 |
|  | 20 | 59.6±0.7 | 65.1±0.6 |
| CUB200 | 5 | 67.1±0.8 | 72.4±0.7 |
|  | 10 | 65.3±0.6 | 69.6±0.6 |
|  | 20 | 64.2±0.5 | 67.8±0.5 |

These results confirm that our dual-space distillation framework maintains stable performance across multiple independent runs and task configurations. The narrow performance gap between the Last Task Accuracy and Average Accuracy further validates the model's robustness and consistent learning behavior throughout the continual process.

## A.5 Computational Cost and Feasibility of LMM Integration

We further analyze the computational overhead introduced by integrating the LMM component from two perspectives: (1) the time and memory cost when querying GPT-4-turbo via API, and (2) the feasibility of fully local deployment using an open-source large multimodal model (LMM).

**Time and Memory Cost with GPT-4-turbo (API-based setup)**

In our experiments, semantic descriptions were queried from GPT-4-turbo for a small number of representative samples per class ($m \in \{0, 1, 2, 3, 4, 5\}$). Here, $m = 0$ denotes the baseline setting without any LMM involvement, where only the visual encoder and adapter are used for forward propagation. We measured training time, memory usage, and the total number of tokens queried on the ImageNet-R dataset under a 5-task incremental setting.

Table 7: Resource usage when querying GPT-4-turbo during training. Bold values denote model training resources, and italic values denote the LMM querying phase.

| $m$ | Train Memory (MB) | Train Time (s) | Tokens Queried | Test Accuracy (%) |
|---|---|---|---|---|
| 0 | 3144 | **3912.47** | — | 50.4 |
| 1 | 3472 | **4836.49** + *1809 (236,053)* | 236K | 81.0 |
| 2 | 3474 | **4931.53** + *4953 (444,700)* | 445K | 81.4 |
| 3 | 3477 | **4941.75** + *7203 (681,405)* | 681K | 81.9 |
| 4 | 3478 | **4939.51** + *8553 (931,251)* | 931K | 82.0 |
| 5 | 3481 | **4998.79** + *9455 (1,158,350)* | 1.16M | 82.2 |

All GPT-4-turbo queries were performed only during training, not inference. As shown in Table 7, increasing the number of representative samples $m$ slightly extends the total training time, but the additional cost remains moderate relative to the performance gain. Once trained, the inference pipeline no longer involves any LMM queries and thus maintains the same efficiency as a standard ViT-based model.

**Local Deployment with an Open-Source LMM**

To evaluate the feasibility of complete local deployment, we replaced GPT-4-turbo with the open-source model Qwen2.5-VL-32B-Instruct-FP8-Dynamic (via Hugging Face). The model was deployed on four A6000 GPUs using FP8 precision, and the same semantic querying and pseudo-label generation pipeline was executed offline.

Table 8: Resource usage when using a locally deployed LMM (Qwen2.5-VL-32B-Instruct-FP8-Dynamic).

| $m$ | Train Memory (MB) | Train Time (s) | LMM Overhead (MB) | Test Accuracy (%) |
|---|---|---|---|---|
| 0 | 3144 | **3915.30** | — | 50.4 |
| 1 | 3478 | **4850.23** + *1483* | 50,024 | 79.6 |
| 2 | 3481 | **4940.17** + *2721* | 50,024 | 79.2 |
| 3 | 3485 | **4952.31** + *4140* | 50,024 | 80.4 |
| 4 | 3489 | **4947.95** + *5946* | 50,024 | 80.6 |
| 5 | 3493 | **5004.80** + *7213* | 50,024 | 80.6 |

The total memory footprint of the locally deployed Qwen2.5-VL model was approximately 50 GB across four GPUs. Compared with GPT-4-turbo, the accuracy decreased slightly (by about 1–1.5%), but the improvement over the baseline without external textual knowledge ($m = 0$) remained substantial. Local deployment introduces additional memory usage only during training, while the inference phase remains identical to a pure visual model. This confirms that full local deployment is feasible without dependence on external APIs or network access.

**Summary**

Overall, the above results demonstrate that integrating an LMM introduces only moderate computational overhead, confined to the training stage. Local deployment is fully feasible on high-memory GPUs (e.g., 4×A6000), and querying a small number of representative samples ($m \leq 3$) provides an effective balance between accuracy and resource efficiency. The proposed framework thus remains lightweight and practical for both API-based and offline multimodal setups.

