# OpenReview forum: "Dual-Space Semantic Synergy Distillation for Continual Learning of Unlabeled Streams"
_NeurIPS.cc/2025/Conference — NeurIPS 2025 poster_

### Official Review · Reviewer_MwSA · 2025-06-05

**Clarity:** 3
**Significance:** 3
**Originality:** 3
**Rating:** 5
**Confidence:** 4

**Summary:**

In this work, the authors propose a method for Unsupervised Continual Learning. They use a dual-space semantic approach where the vision component is processed through a CLIP ViT-L/14 encoder, and GPT-4-turbo processes the text. The approach consists of extracting the visual embeddings of the samples in the current task and creating clusters. The most significative images (i.e., the top-k samples closest to each centroid) are then used to query GPT-4-turbo for preliminary predictions. In addition to the image, the LMM is prompted with 3 different prompts to have different levels of granularity. These two embeddings are merged together to create pseudo-labels based on the confidence of the predictions of the two spaces. After the pseudo-labeling, an adapter for the visual encoder is trained using an objective considering the agreement between the LMM and the visual encoder. Finally, they also add a knowledge distillation component to reduce forgetting. The authors evaluate their approach on ImageNet-R, CIFAR100, and CUB200 with various classes per task. The evaluation comprises an ablation of all the components in the objective function. In the supplementary materials, the authors perform further ablation and analysis.

**Questions:**

- How do the methods compare in a fair environment with the same backbone?

- Does the LMM introduce a noticeable overhead regarding time and memory requirements?

**Ethical Concerns:**

["NO or VERY MINOR ethics concerns only"]

**Final Justification:**

My initial concerns were primarily due to unclear reporting of backbone consistency and the lack of analysis on LMM-related computational overhead. The authors have now provided thorough clarifications and additional experiments addressing both points, including detailed evaluations of API-based and local deployments. The method is well-motivated, technically sound, and demonstrates strong empirical results. Based on these improvements, I am updating my rating to Accept.

**Limitations:**

yes

**Quality:**

3

**Strengths And Weaknesses:**

## Strengths

- The manuscript is straightforward and easy to read.

- The pipeline is interesting and smartly introduces LMMs in the whole framework.

- The results are really promising.

- The ablations are exhaustive and confirm the importance of each component.

## Weaknesses

- The results of the competitors reported in Tab. 1 employ different backbones with respect to the pretrained ViT-L/14 used by the authors. Indeed, the numbers are extremely close to those reported by the respective papers. Specifically, MSc-iNCD employs a ViT-B/16,  while POCON, PFR, and CassLE use a ResNet-18. In addition to the fact that the authors also use GPT-4-turbo (while the number of parameters of the turbo version is not public, GPT-4 has around 1.7 trillion parameters), this results in a very unfair comparison. The authors should use the same visual backbone for all methods and better report the competitors' backbone.

- A computational analysis and comparison are missing, which is important given that the employed LMM is not negligible.

---

> ### Author Rebuttal · Authors · 2025-07-31
>
> We thank Reviewer MwSA for recognizing the strengths of our work, particularly the effective integration of multimodal information under the UCIL setting and the comprehensive ablation study validating each module. We appreciate your concerns regarding backbone consistency and the computational overhead of LMMs, which we address in detail below. We will further clarify these aspects in the revised manuscript and release the complete code to the community. Thank you for your constructive feedback and we look forward to your updated rating.
>
> ## Clarification on Backbone Consistency Across Methods in Table 1(Questions 1)
>
> All results in Table 1 have already been reproduced using the same visual backbone **ViT-L/14**. We apologize that this was not clearly stated in the main paper, which may have caused confusion. We will revise the manuscript to explicitly specify the backbone used for each baseline and add a note below the table indicating that **all results are reproduced with ViT-L/14** for fair comparison. We are committed to making the code publicly available for community research in the final version.
>
>
>
> ## Time and Memory Overhead of Using LMMs(Questions 2)
>
> **Unsupervised Class-Incremental Learning (UCIL)** is a highly practical yet extremely challenging setting, primarily due to the complete absence of ground-truth labels throughout the continual learning process. With the emergence of large language models (LLMs), leveraging their semantic prior knowledge as a form of supervision has become a promising direction. However, in our experiments conducted on the five-stage incremental setting of ImageNet-R (as described in Section 4.2), we observed that this strategy incurs **significant computational overhead** without delivering corresponding performance gains.
>
> Specifically, we experimented with using LLMs to generate labels for **every single sample**, which were then used for text-based clustering (**Text Clustering**) and downstream model training. Even when combined with visual feature alignment (**Prototype + Language KD**), this approach achieved only **76.8%** accuracy. In contrast, our proposed strategy selects only a **few representative samples per class** (e.g., 3 samples) for LLM queries to construct class-level semantic prototypes. This dramatically reduces the computational cost while yielding better performance. Moreover, when combined with our **proposed Semantic Alignment Distillation (SAD)** mechanism, the method achieves the best results.
>
> | Text Clustering Only | Prototype + Language KD | Visual + Prototype + Language KD | Full Method (Ours) |
> | :------------------: | :---------------------: | :------------------------------: | :----------------: |
> |         51.1         |          76.8           |               78.1               |      **81.9**      |
>
>
>
> These results clearly demonstrate that **our method not only alleviates the time and memory overhead brought by LLMs, but also achieves superior classification performance**. This strategy—**selective LMM integration combined with cross-modal collaborative supervision**—strikes a favorable balance between efficiency and effectiveness, showing strong practicality and scalability in real-world UCIL scenarios.

---

> > ### Comment · Reviewer_MwSA · 2025-08-01
> >
> > Thank you to the authors for the response. However, the concern regarding time and memory consumption during inference has not been fully addressed.
> >
> > Specifically, my question regards to the additional computational overhead introduced by incorporating the LMM component into the pipeline. I am interested in the difference in time and memory usage between the two setups:
> >
> > 1. Performing a forward pass through only the visual encoder and adapter, and
> >
> > 2. Performing a forward pass that also includes the LMM with various prompts applied to the image output.
> >
> >
> > Could the authors clarify the delta in terms of inference time and memory usage between these two approaches?
> >
> > Additionally, is it feasible to run the full pipeline locally? If the LMM component is accessed via API, could you provide an estimate of the average cost per sample for inference?

---

> > > ### Author Response · Authors · 2025-08-02
> > > **Clarifying Time and Memory Overhead of LMM Usage**
> > >
> > > We thank the reviewer for raising a more specific question. In response, we provide detailed empirical results from two perspectives: (1) the time and memory cost using the GPT-4-turbo API, and (2) the feasibility and resource profile of local deployment using an open-source multimodal model.
> > >
> > > ### **1. Time and Memory Cost with GPT-4-turbo (API-based setup)**
> > >
> > > In our original experiments, we queried semantic descriptions from GPT-4 Turbo via the API for a small number of representative samples per class. To quantify the associated resource overhead more clearly, we conducted systematic measurements on the ImageNet-R dataset under a 5-task incremental setting, varying the number of queried samples per class m∈{0,1,2,3,4,5}, where m=0 represents the forward pass only has the visual encoder and adapter.
> > >
> > > **Table 1: Resource usage with GPT-4-turbo**
> > >
> > > |  m   | Train Memory (MB) |              Train Time (s)               | Test Accuracy (%) | Test Memory Usage（MB） |
> > > | :--: | :---------------: | :---------------------------------------: | :---------------: | :---------------------: |
> > > |  0   |     **3144**      |             **3912.47** + *0*             |       50.4        |        **3144**         |
> > > |  1   |     **3472**      |  **4836.49** + *1809* (*236,053 tokens*)  |       81.0        |        **3472**         |
> > > |  2   |     **3474**      |  **4931.53** + *4953* (*444,700 tokens*)  |       81.4        |        **3474**         |
> > > |  3   |     **3477**      |  **4941.75** + *7203* (*681,405 tokens*)  |       81.9        |        **3477**         |
> > > |  4   |     **3478**      |  **4939.51** + *8553* (*931,251 tokens*)  |       82.0        |        **3478**         |
> > > |  5   |     **3481**      | **4998.79** + *9455* (*1,158,350 tokens*) |       82.2        |        **3481**         |
> > >
> > > In this table, **bold** numbers indicate training time and memory; *italic* numbers refer to the LMM querying phase using GPT-4-turbo. Notes:
> > >
> > > - We report the total number of tokens consumed by the LMM queries under each value of m;
> > > - All LMM queries are performed at the model training phase; **the test inference stage does not involve the LMM at all once the model is already trained**. Based on the test results, we believe that the additional training cost is entirely worthwhile, as it can lead to a significant improvement in the model's performance at a relatively low cost.
> > >
> > > ### **2. Feasibility and Resource Profile of Local LMM Deployment**
> > >
> > > To further address the reviewer's question about full local deployment, we experimented with replacing GPT-4-turbo using an open-source LMM—**Qwen2.5-VL-32B-Instruct-FP8-Dynamic** (via Hugging Face). The model was deployed on 4 GPUs using FP8 precision, and we ran the semantic querying and label generation pipeline **offline**.
> > >
> > > **Table 2: Results with local LMM (Qwen2.5-VL)**
> > >
> > > |  m   | Train Memory Usage (MB) |    Train Time (s)    | Test Accuracy (%) | Test Memory Usage (MB) |
> > > | :--: | :---------------------: | :------------------: | :---------------: | :--------------------: |
> > > |  0   |     **3144** + *0*      |  **3915.30** + *0*   |       50.4        |        **3144**        |
> > > |  1   |   **3478** + *50024*    | **4850.23** + *1483* |       79.6        |        **3478**        |
> > > |  2   |   **3481** + *50024*    | **4940.17** + *2721* |       79.2        |        **3481**        |
> > > |  3   |   **3485** + *50024*    | **4952.31** + *4140* |       80.4        |        **3485**        |
> > > |  4   |   **3489** + *50024*    | **4947.95** + *5946* |       80.6        |        **3489**        |
> > > |  5   |   **3493** + *50024*    | **5004.80** + *7213* |       80.6        |        **3493**        |
> > >
> > > Clarifications:
> > >
> > > - **Bold** values indicate model training memory and time; *italic* values refer to the additional resource consumption from local LMM querying;
> > > - The total LMM memory footprint was approximately **50024 MB** across 4 GPUs(A6000 * 4);
> > > - Compared to GPT-4-turbo, the accuracy dropped slightly (~1–1.5%), however, compared with no external text knowledge introduced (m=0), the performance is still greatly improved.
> > >
> > > These results demonstrate that **entirely local deployment is feasible**, with no dependence on APIs or internet connection. Although local deployment introduces some additional memory overhead, it is limited to the **initial stage training**, and **has no impact on the inference pipeline**, which remains identical to a standard ViT-based model.
> > >
> > > Moreover, our method is carefully designed to **query only a few representative samples per class**, which not only reduces resource consumption but also **eases the risk of text query noise**, making it suited for open-source multimodal models. As open-source pre-trained models continue to develop, our approach will have the opportunity to achieve better applicability. We will include a full report on local deployment performance in the final version of the paper.

---

> ### Author Response · Authors · 2025-08-05
>
> We sincerely thank Reviewer MwSA again for the detailed follow-up question and for giving us the opportunity to clarify the resource implications of integrating LMMs into our pipeline. Following your request, we conducted extensive additional experiments and provided both API-based (GPT-4-turbo) and fully local (Qwen2.5-VL-32B) deployment results, reporting inference memory usage, latency, and token-level breakdown across different settings of queried samples (m = 0 to 5). We also explained the feasibility and stability of the local setup and its relevance to real-world UCIL applications.
>
> We hope these clarifications have fully addressed your concerns. If any part remains unclear or further elaboration is needed, we would be very happy to provide additional details.
>
> Your expert feedback is of great importance to us. If possible, we would deeply appreciate it if you could let us know whether our responses have sufficiently resolved your questions. Thank you again for your valuable time and thoughtful comments.

---

> > ### Comment · Reviewer_MwSA · 2025-08-05
> >
> > Thank you for the detailed and thorough clarifications, both regarding the consistency of the backbone and the time/memory overhead of LMM integration. I appreciate the additional experiments, especially the breakdown between API-based and fully local deployments, which provide further details on the method’s practical feasibility and design. These additions strengthen the manuscript and address my original concerns.
> >
> > I will update my rating to Accept to reflect these improvements.

---

### Official Review · Reviewer_Mfes · 2025-06-20

**Clarity:** 4
**Significance:** 3
**Originality:** 3
**Rating:** 5
**Confidence:** 3

**Summary:**

This paper uses a multimodal approach to do unsupervised class incremental learning. It uses the visual embeddings initially, then textual descriptors on cluster centers, to help guide cluster annotation. The results look pretty good.

**Questions:**

All already asked

**Ethical Concerns:**

["NO or VERY MINOR ethics concerns only"]

**Final Justification:**

The overall positive review need not change

**Limitations:**

yes

**Quality:**

3

**Strengths And Weaknesses:**

I personally felt this is one of the better written papers I've seen in this area; the whole story was very clear and well motivated. The results also look very good. I have some comments that could maybe be considered for improving the paper, and I have to admit I'm not familiar with related work so I can't comment on novelty, but for the most part the paper seems a solid contribution to me.

weaknesses / areas for improvement:
 - Sometimes the language is a bit precise. For example, using "noisy" to describe the visual embedding system feels imprecise; after all, it's not the images or the labels that are noisy. I guess it's something like, a lizard and gecko may look alike, but the image embedding doesn't always discriminate at that precise level as, say, the orientation of the scales or distance of the eyes. This seems more like a lack of focused or intentional granualiry, rather than noise.

 - in the abstract, what is "semantic alignment distillation"? Is this a commonly used thing?
 - Why is language more stable than vision? Is this a feature / fault of the model, or of the data distribution, or something else?

 - line 70: biases in labels cannot intorduce *stochastic* noise. In general I think the use of words like "randomness", "noise", and "stochastic" should be more carefully doled out; if something is inherent in the task or dataset, it cannot be random. See also line 187, line 223.

 - Minor: 162: "in the figure" which figure?

 - minor: 235: could be more explicit that C^t isn't an exponent, but you compute this for all j = 1,...,C^t and then pick the top 1,2. Maybe describe (6) in two lines?

 - (11),(12) How sensitive is this to the choice of lambda1, lambda2? I guess drift could be a big issue?

 - line 287: do you mean after training, or during training? Do you evaluate during the sequence, or all at once after the sequence is doled out?

 - imprecise: line 319 "This performance is attributed to the low resolution of CIFAR images...." But this is true for all the other baselines, no? The more precise the language, especially in experimental results, the better.

---

> ### Author Rebuttal · Authors · 2025-07-31
>
> We sincerely appreciate Reviewer Mfes for the encouraging feedback, especially the recognition of the clarity and potential of our methodology and manuscript presentation. Your comments have helped us identify areas where the language, terminology, and experimental explanations can be further refined. We address each of your points below and will revise the manuscript accordingly to improve precision and clarity. We also plan to release our code and models upon publication to ensure reproducibility. Thank you again for your support and valuable suggestions.
>
> ## Clarification on the Use of the Term “Noisy” (Improvement 1)
>
> We sincerely thank the reviewer for the insightful comment. You are absolutely right that the term “noisy” used to describe the visual embedding system may be imprecise. Our original intent was to address the inaccuracy of the generated pseudo labels, which arise when visual features fail to distinguish between fine-grained categories fully. In the literature on unsupervised and noisy-label learning, this phenomenon is often referred to as the label noise problem [1].
>
> We greatly appreciate your suggestion, and in the revised version, we will adopt more precise terminology, such as “sub-optimally discriminative representations” or “lack of fine-grained semantic discrimination.” We will also clarify in context that our concern lies with the discrepancy between pseudo labels and ground-truth semantics, not with stochastic noise in the feature extraction process itself.
>
> **Reference:**
>
> [1] Han, B., Yao, Q., Yu, X., Niu, G., Xu, M., Hu, W., Tsang, I. W., & Sugiyama, M. (2020). _A Survey of Label-noise Representation Learning: Past, Present and Future_. arXiv preprint arXiv:2011.04406.
>
>
>
> ## Clarification on “Semantic Alignment Distillation” (Improvement 2)
>
> Thank you for pointing out the ambiguity regarding the term “semantic alignment distillation.”
>
> We acknowledge that **“semantic alignment distillation” is not a standard term in the literature**, but rather a novel distillation strategy proposed in this paper. The core idea is to **leverage the stability of the semantic space (i.e., the text encoder representations) to align and constrain the visual feature space**, thereby mitigating catastrophic forgetting in the class-incremental learning process.
>
> Concretely, after training on each task, we preserve the semantic prototypes and covariance matrices derived from pseudo textual labels. During subsequent tasks, we use these saved semantic prototypes to guide the model such that visual features of old-class samples remain close to their corresponding semantic descriptions while being **repelled from those of new classes**. This contrasts with traditional distillation approaches (e.g., logit-based or feature matching), and instead **introduces a cross-modal (semantic-to-visual) constraint**.
>
> We chose the term _semantic alignment distillation_ to emphasize two aspects:
>
> 1. **Semantic**: The targets for distillation originate from the **semantically stable text space**, not from the evolving visual encoder.
>
> 2. **Alignment distillation**: The goal is to **align visual representations with previously acquired semantic knowledge** across tasks, ensuring continuity in the feature space despite the incremental setup.
>
>
>
>  ## The Stability of the Language Space Compared to the Visual Space (Improvement 3)
>
> We thank the reviewer for raising this important point. Our claim that the **language (text) space is more stable than the visual space** is based on both theoretical reasoning and empirical design:
>
> 1. **Frozen representations**: Textual features generated by the LMM (GPT-4-turbo) remain frozen during training, providing consistent semantic anchors across tasks. In contrast, visual features are updated via task-specific adapters, making them more prone to drift.
> 2. **Category-level abstraction**: Language naturally encodes semantic concepts at the class level and is less sensitive to variations such as pose or background, offering more stable and generalizable representations.
> 3. **Temporal consistency**: While visual distributions evolve with each new task, textual prototypes—built from a few representative samples—remain unchanged, helping to preserve semantic continuity.
>
> These properties justify our use of the text space as a stable reference for semantic alignment across incremental tasks.
>
>
>
> ##  The Use of the Term “Stochastic Noise” vs. “Bias” (Improvement 4)
>
> We thank the reviewer for pointing out the imprecise use of “stochastic noise.” We agree that **biases inherent to the data should not be described as random**. Our original intent was to express that **semantic ambiguity and bias lead to label inconsistency**, not true randomness.
>
> To address this, in the revised version:
>
> 1. We will replace “stochastic noise” with more accurate terms such as “label uncertainty” or “pseudo-label inconsistency.”
>
> 2. We will add clarifications to indicate that “noise” refers to semantic misalignment or unreliability in pseudo-labels, not statistical perturbations.
>
> We appreciate the reviewer’s careful reading and will revise the manuscript accordingly to improve clarity and rigor.
>
> ## Clarification of Figure Reference in Line 162(Improvement 5)
>
> This refers to our system framework, which is shown in **Figure 3**. Thank you for pointing this out. We will clarify the reference in the final version.
>
> ## Clarification of Formula and Notation for C^t in Line 235 (Improvement 6)
>
> We thank the reviewer for pointing out the ambiguity in our notation involving C^t in line 235.
>
> In our original writing, C^t was intended to denote the **number of classes in the current task**, but we recognize that the superscript form could be easily misinterpreted as an exponent. Moreover, the compact expression we used may have hindered interpretability.
>
> In the revised version, we will take the following actions to address this issue:
>
> 1. **Explicitly clarify that C^t denotes the number of classes**, not an exponent.
> 2. **Rephrase the original expression into clearer steps**, improving readability and interpretability of our confidence-based weighting mechanism.
>
> We believe these revisions will eliminate confusion and enhance clarity. We appreciate the reviewer’s insightful and constructive feedback.
>
> ## The Sensitivity of Hyperparameters $\lambda_1$ and $\lambda_2$ (Improvement 7)
>
> We thank the reviewer for highlighting the importance of hyperparameter sensitivity. We have conducted a sensitivity analysis of the hyperparameters $\lambda_1$ and $\lambda_2$, and the corresponding results are included in the supplementary material.
>
>
>
> ## Clarification of Evaluation Strategy and Metrics (Improvement 8)
>
> Thank you for your question regarding the evaluation protocol. In our main experiments, we adopt the standard **last-stage evaluation strategy**, where the model is evaluated **after completing all tasks** on **the entire set of classes** learned so far. Accordingly, the main metric we report in the paper is **Last Stage Accuracy (LA)** — a commonly used measure in unsupervised class-incremental learning (UCIL) to assess final performance.
>
>
>
> ## Clarification on the Impact of Image Resolution on Method Comparison(Improvement 9)
>
> We agree with your point: **all methods are evaluated at the same image resolution**, so we cannot simply attribute the lower resolution as a unique disadvantage of our method. Our intention was not to emphasize that "low resolution leads to poorer performance in our method," but rather to highlight that, **compared to datasets with higher resolutions (such as ImageNet-R and CUB200)**, generating LMM-based semantic descriptions on CIFAR-100 is more affected by image quality limitations, which indeed impacts the extraction and discrimination of semantic information to a greater extent.
>
> In other words, the key component of our method—**the fine-grained semantic description capability of the LMM**—is less stable and effective in the CIFAR-100 setting compared to higher-resolution datasets. This has a greater impact on the quality of the pseudo-labels generated by our method. To avoid misunderstanding, we will clarify this statement in the revision and more accurately emphasize that the issue arises from **the higher dependency of our method on the semantic information perceived from the images**, rather than the dataset itself.

---

> > ### Comment · Reviewer_Mfes · 2025-08-04
> > **Thanks**
> >
> > Thanks for the rebuttal. The responses were very interesting to read, and I agree with the choices for changes that you suggest. I still wonder if there is an air of subjectiveness to the claim that language is more stable than images; it seems I could cook up a story about how images don't change, but language changes person to person, or something, but the explanation you gave is pretty convincing too -- just feels subjective overall. Overall, I maintain my score.

---

> ### Author Response · Authors · 2025-08-05
>
> We sincerely thank Reviewer Mfes for the thoughtful and encouraging feedback.
>
> Your comments greatly helped us refine our terminology and improve the clarity of our presentation. Regarding the discussion on the stability of language versus vision, we appreciate your insightful perspective and fully understand that different interpretations are possible depending on context.
>
> While we will carefully consider your point of view, we also continue to build on our core intuition—that textual features produced by a frozen language model can offer a stable semantic reference across tasks, particularly in the UCIL setting. We will strive to present this argument more clearly and precisely in the final version, making our assumptions and motivations transparent.
>
> Thank you again for your valuable input and supportive evaluation.

---

### Official Review · Reviewer_TKos · 2025-06-30

**Clarity:** 3
**Significance:** 2
**Originality:** 2
**Rating:** 4
**Confidence:** 5

**Summary:**

This paper focuses on class incremental learning of vision-language models (VLMs). While existing approaches have achieved increasing success in supervised class incremental learning, they struggle to deal with complex visual scenarios and fully unsupervised settings that appear in many real applications. This is because the LLMs capture nuanced distinctions among visually similar samples, and inherent knowledge biases and inter-category semantic ambiguity introduce substantial stochastic noise in pseudo labels. To address this limitation, this work designs a hybrid approach that leverages the coarse-to-fine characteristic of language modality to refine the visual samples and also proposes a language-guided collaborative alignment to improve visual feature discrimination. The main contributions of this work are three-fold: (1) proposing a low-resource UCIL that captures the hierarchical semantic description to enhance the semantic supervision quality of unlabeled samples, (2) exploiting dual-modality pseudo-labeling strategy that jointly leverages visual and semantic cues for robust representation learning, and (3) conducting comprehensive experiments that demonstrate the efficiency of the proposed methods.

**Questions:**

1. For a fair comparison, the compared methods designed for CLIP and methods using LLMs in CIL should be reported in an unsupervised manner. The comparison experiments of CLIP-based class incremental learning methods, such as ZS-CLIP [2], MOEAdapeter [3], and RAPF [4] are not sufficient. Methods with LLMs, such as GMM [5] should also be compared.

2. It has been proved that the impressive vision-text zero-shot ability of the CLIP in ZS-CLIP [2]. Given the coarse hypothesis that the origin text label is the upper bound performance, the pseudo label quality of pure LLMs and the proposed methods should be compared with the real text labels for precise verification.

[1]Sylvestre-Alvise Rebuffi, Alexander Kolesnikov, Georg Sperl, and Christoph H Lampert. icarl: Incremental classifier and representation learning. In CVPR, pages 2001–2010, 2017.

[2]Alec Radford, Jong Wook Kim, Chris Hallacy, Aditya Ramesh, Gabriel Goh, Sandhini Agarwal, Girish Sastry, Amanda Askell, Pamela Mishkin, Jack Clark, et al. Learning transferable visual models from natural language supervision. In ICML, pages 8748–8763, 2021.

[3]Zhou, D.W., Ye, H.J., Zhan, D.C., Liu, Z.: Revisiting class-incremental learning with pre-trained models: Generalizability and adaptivity are all you need.

[4]Linlan Huang, Xusheng Cao, Haori Lu, and Xialei Liu. Class-incremental learning with clip: Adaptive representation adjustment and parameter fusion. In ECCV, 2024.

[5]Xusheng Cao, Haori Lu, Linlan Huang, Xialei Liu, Ming-Ming Cheng.  Generative Multi-modal Models are Good Class-Incremental Learners. In CVPR, 2024.

**Ethical Concerns:**

["NO or VERY MINOR ethics concerns only"]

**Final Justification:**

Thanks for addressing all my concerns. The rebuttal is convincing. I will raise my rating.

**Limitations:**

yes

**Quality:**

3

**Strengths And Weaknesses:**

Strength

The idea of exploring robust and accurate semantic supervision of unlabeled samples based on the coarse-to-fine descriptions of LLMs and complementary strengths of visual and text information is instructive. The entire framework of constructing the hypergraph and designing the hypergraph convolution is well-explained. The experiments' analysis depicts the effectiveness of the proposed approach.

Weakness

However, some aspects of the manuscript and experiment results need further clarification and support. Below, I provide some minor and major comments.

1. The novelty of the proposed method is relatively low in solving the main challenge of unsupervised CIL. The pseudo-labeling technique relies on the LLMs. The main contribution of preventing catastrophic forgetting, corresponding to section 3.3 is also not novel. The metric learning loss in Eq. (10) and distillation loss in Eq. (11) have been broadly used in CIL and unsupervised CIL methods.

2. Page 5, line 194-195, how to determine the most representative m samples from each clustering result?

3. Provide the computation method of the evaluation metric proposed in Section 4.1 . It is recommended to report the results of two metrics, the last stage average accuracy and overall average accuracy following [1], which have been broadly utilized in CIL papers, in the experiment part to validate the performance superiority.

4. Page 6, line 234, the symbol of $i$ should not be subscript.

---

> ### Author Rebuttal · Authors · 2025-07-31
>
> We thank Reviewer TKos for recognizing the merits of our work, especially your positive comments on our framework that leverages language modality to enhance semantic supervision under the UCIL setting. We acknowledge your suggestions for clarifying the novelty, reporting evaluation metrics, and improving the presentation in certain areas. We respond to each of your comments in detail below, and will include the necessary theoretical discussions and experimental updates in the revised manuscript. We will release the source code to the community and hope that the Reviewer will reconsider the rating of our paper.
>
>
> ## Novelty(Weakness1)
>
> We thank the reviewer for raising this important point. We clarify that the core novelty of our work **lies not in merely calling LLMs**, but in **how we systematically integrate their semantic priors into a closed-loop pseudo-labeling and fusion framework to enable vision-language collaborative supervision**.
>
> To our knowledge, **this is the first work to incorporate LLMs under the challenging Unsupervised Class-Incremental Learning (UCIL) setting**, where no real labels are available. We avoid querying LLMs for every sample due to inefficiency and noise, as demonstrated in Figures 2 and 5. Instead, we query only a few representative samples per class (e.g., 3), ensuring semantic stability and low overhead. More critically, we introduce a **cross-modal co-supervision mechanism** that combines visual sensitivity with language consistency to enhance learning.
>
> Regarding Section 3.3 and the SAD module, we would like to clarify that SAD is **not a simple extension of traditional prototype-based methods**, but rather a **cross-modal alignment strategy that leverages the stability of the language space to guide the evolution of the visual space**.
>
> Our proposed **hierarchical prompt template** generates high-quality and consistent class descriptions, forming a set of textual prototypes with strong discriminability and structural stability. Since the text encoder remains frozen during training, the distribution of these features remains unchanged throughout the incremental learning process. We therefore use the relative distances between textual prototypes as a set of **stable anchors**, which are projected into the visual space to constrain its drift and mitigate catastrophic forgetting.
>
> Specifically, SAD constructs a directional distillation loss (Eq. 10) by comparing each old sample’s distance to its original text prototype and to new class prototypes, thereby enforcing structural alignment between the language and visual spaces.
>
> Although SAD may appear superficially similar to certain prototype-based distillation methods in form (e.g., using class prototypes as constraints), **the fundamental design philosophy is distinctly different**. Traditional approaches typically construct prototypes in the visual space and focus on preserving or reconstructing visual feature structures. In contrast, our method **transfers semantic structure from the language space to the visual space via cross-modal alignment**. Finally, our ablation results in Table 2 (rows d → e) empirically demonstrate the significant contribution of SAD to overall performance, validating both its effectiveness and independent value.
>
> ## Selection of Representative Samples(Weakness2)
>
> Our goal is to achieve ideal recognition performance with a small text supervision cost. We have detailed the representative sample selection process in lines 192–197 on page 5. Specifically, we apply K-means clustering to obtain class centers, then select the top-m samples with the highest cosine similarity to each center. This ensures strong visual coherence and supports the generation of high-quality text descriptions.
>
> Additionally, we analyze the impact of the parameter _m_ in the supplementary material. As shown in Table 1, increasing _m_ from 1 to 5 improves classification accuracy from 81.0% to 82.2%, suggesting that selecting more representative samples enhances the semantic richness of pseudo-labels.
>
> ##  Metrics(Weakness3)
>
> Following the reviewer’s suggestion, we have re-evaluated the performance of our method on the ImageNet-R dataset under three different task splits (5-task, 10-task, and 20-task) based on the setting in Section 4.1 of the paper. We now report two more standard evaluation metrics commonly used in class-incremental learning: **Last Task Accuracy** and **Average Accuracy**, as shown in the tables below. Results for the other datasets (e.g., CIFAR-100 and CUB-200) will be updated accordingly in the revised version.
>
> | Tasks | Last Task Accuracy (%) | Average Accuracy (%) |
> | :---: | :--------------------: | :------------------: |
> |   5   |          81.9          |         89.9         |
> |  10   |          82.3          |         86.4         |
> |  20   |          79.4          |         83.2         |
>
> ## Format(Weakness4)
>
> Thank you for the careful review. We confirm that the subscript formatting of the variable _i_ in line 234 is indeed inconsistent. We will correct this in the final version to ensure consistency and proper notation throughout the paper.
>
> ## Inapplicability of Fully-Supervised Baselines under the UCIL Setting(Question1)
>
> We appreciate the Reviewer’s concern regarding the fairness of comparisons. To begin, we would like to clarify the **fundamental differences between Unsupervised Class-Incremental Learning (UCIL)** and traditional supervised CIL, which are essential for properly contextualizing the scope of our method.
>
> In the UCIL setting, the model has **no access to any ground-truth class labels throughout training**. At each stage, it receives only unlabeled images from novel classes. In most cases, the number of new classes is known, but their specific identities are not. The model must rely on pseudo-labeling mechanisms and self-supervised structures to gradually construct semantic knowledge—introducing uncertainty and openness far beyond that of traditional supervised setups. Prior work, such as CURL [1] and UCIL-Confusion [2], has demonstrated that methods relying on label supervision are generally unsuitable in this context. Furthermore, the classic clustering survey by Jain et al. [3] emphasizes that in unsupervised clustering, the class structure must be discovered solely based on inter-sample similarity, making the task fundamentally different from supervised learning.
>
> Under this premise, we have carefully reviewed the methods highlighted by the Reviewer—RAPF, ZS-CLIP, MOEAdapter, and GMM—and based on their design assumptions and workflow, we argue that they cannot be fairly compared under the UCIL setting for the following reasons:
>
> 1. **RAPF** relies on ground-truth labels to construct semantic similarity constraints and perform feature distillation and parameter fusion. In UCIL, we can only use pseudo-labels derived from visual clustering or LMM queries. Our preliminary experiments show that these pseudo-labels are of low quality, yielding classification accuracies of only 36.9% (visual space) and 51.1% (language space), which are insufficient to support RAPF's supervision-driven mechanisms.
> 2. **ZS-CLIP** assumes access to known class names to construct text prototypes for zero-shot inference. However, class names are unknown in UCIL, making its assumptions incompatible with ours.
> 3. **MOEAdapter** requires explicit class labels at each stage to activate expert routing and build supervised prototypes. These components cannot function without supervision and thus are not transferable to UCIL.
> 4. **GMM**, though incorporating a language module, relies heavily on class labels for pseudo-label modeling, contrastive learning, and text alignment. When labels are unavailable, their core mechanisms fail to operate and may even collapse due to semantic misalignment.
>
> In summary, while these methods are representative under their respective **supervised** settings, they are all fundamentally dependent on labeled data for both training and structural design. This makes them incompatible with the **fully unsupervised UCIL** framework we propose. Therefore, including them in direct comparison under the UCIL setting would be theoretically inappropriate and practically infeasible. Instead, our experiments focus on comparisons with established UCIL methods and demonstrate the robustness and effectiveness of our approach on benchmarks such as ImageNet-R.
>
> **Reference:**
>
> [1] Rao, D. Continual Unsupervised Representation Learning. *Advances in Neural Information Processing Systems.
>
> [2] Gautam, R. Unsupervised Class-Incremental Learning Through Confusion. *arXiv preprint arXiv:2110.06674*.
>
> [3] Jain, A. K. Data clustering: A review. *ACM Computing Surveys (CSUR)*, 31(3), 264–323.
>
> ## Verify Text Label Quality(Question2)
>
> We thank the reviewer for the valuable suggestion. To evaluate pseudo-label quality, we conducted three experiments on ImageNet-R with five incremental tasks:
>  (1) In an open-world setting, the LLM generates diverse, unstructured labels for each sample. To address this, we applied clustering to the generated texts and utilized the cluster centers as class prototypes, achieving an accuracy of 76.8%.
>  (2) Under the same unsupervised assumption (unknown classes), our proposed hierarchical prompting queried only a few representative samples per class to construct consistent and discriminative prototypes, improving performance to 81.9%.
>  (3) Using ground-truth labels to construct “A photo of [label]” text prompts served as a supervised upper bound (84.2%).
>
> Only (3) uses real label names. Our method, without any true labels, achieves performance close to the upper bound, confirming the effectiveness of our semantic-guided pseudo-labeling design. Detailed experiments will be supplemented in the revised version.

---

> > ### Comment · Reviewer_TKos · 2025-08-08
> >
> > Thanks for addressing all my concerns. The rebuttal is convincing. I will raise my rating.

---

> ### Author Response · Authors · 2025-08-05
>
> We sincerely thank Reviewer TKos again for your insightful feedback, which significantly helped us improve the clarity and rigor of our work. In our previous response, we have addressed all the points you raised in detail, including the clarification of our method’s novelty, the selection process for representative samples, the addition of standard evaluation metrics, and the rationale for excluding fully supervised baselines under the UCIL setting.
>
> We have also provided new experimental results and theoretical justifications to support our claims, and we plan to include all these updates in the revised manuscript. We would be truly grateful if you could let us know whether our responses have resolved your concerns or if any further clarification is needed. Your feedback is extremely valuable to us, and we would be more than happy to continue the discussion if needed.
>
> Thank you again for your thoughtful and constructive comments.

---

> ### Author Response · Authors · 2025-08-07
> **Further Explanation**
>
> Dear Reviewer TKos,
>
> Thank you again for your thoughtful and detailed feedback during the review process. We have carefully addressed each of your comments in both our Rebuttal and subsequent official comment, including additional experiments and clarifications. We would be grateful to know whether you have had a chance to review our responses. If any issues remain unclear, we would be more than happy to provide further clarification.
>
> Here, we would like to offer additional explanation regarding your comment on “verifying the zero-shot capability of CLIP,” particularly in the context of our *Unsupervised Class-Incremental Learning (UCIL)* setting.
>
> As you rightly noted, CLIP has shown impressive zero-shot performance in various **supervised** scenarios. However, these methods typically **require a predefined set of class names**, using prompts like “A photo of a dog” to match against image features. This foundational assumption does not hold in UCIL: in our setting, **new classes arrive incrementally with no access to real labels or class names** throughout training.
>
> Due to the lack of such prior knowledge, standard zero-shot classification strategies with CLIP become inapplicable. Instead, existing UCIL methods typically rely on unsupervised clustering in either the visual or textual feature space. However, as demonstrated in our paper, such clustering is often insufficient to produce reliable pseudo-labels—especially when dealing with semantically ambiguous or visually similar categories.
>
> The methods you mentioned (e.g., RAPF, GMM) incorporate CLIP's zero-shot capabilities in their algorithmic design, which often implicitly rely on certain forms of supervised priors, such as predefined class sets or semantically guided label prompts. While such mechanisms can enhance performance in supervised class-incremental settings, they become inapplicable in our UCIL setting, where no class names are available. As a result, these methods are not directly comparable to ours, either semantically or technically. We will further emphasize this point in the revised manuscript to more clearly delineate the usage and applicability of CLIP under different learning paradigms.
>
> We will make this distinction more explicit in our revised manuscript to clarify the boundaries of CLIP's zero-shot applicability across different learning settings. Once again, thank you for your constructive comments and continued engagement—we greatly appreciate your insights.

---

### Official Review · Reviewer_jsLU · 2025-07-01

**Clarity:** 4
**Significance:** 3
**Originality:** 3
**Rating:** 4
**Confidence:** 5

**Summary:**

This paper focuses on unsupervised class-incremental learning and proposes a novel framework to address its challenges. The method leverages CLIP and GPT-4-turbo to generate more reliable pseudo labels for training data. It introduces both visual pseudo labels and textual pseudo labels to complement each other. Additionally, knowledge distillation is employed to mitigate catastrophic forgetting. Experimental results demonstrate the effectiveness of the proposed approach.

**Questions:**

See weakness.
I will consider to raise my rating if the authors can address all my concerns.

**Ethical Concerns:**

["NO or VERY MINOR ethics concerns only"]

**Final Justification:**

I appreciate the authors' efforts, and their rebuttal has addressed some of my concerns. However, I find the proposed prototype-based method still to be relatively common. As a result, I have decided to raise my score from 2 to 4.

**Limitations:**

yes

**Quality:**

3

**Strengths And Weaknesses:**

Strengths
1.The motivation behind the proposed method is compelling. The authors aim to address the challenge of inaccurate pseudo labels in existing methods, which is a meaningful problem.
2. The design of Prompt 3 within the Semantic Collaborative Facilitative Supervision module is both novel and insightful. It captures common visual scenes associated with objects, which is often overlooked in prior works.
3. The paper is well-written, with high-quality presentation and clear structure.

Weaknesses
1. Novelty of Semantic Alignment Distillation: The authors utilize stored class prototypes and matrices to alleviate forgetting, which aligns with standard practices in prototype-based continual learning. This technique has been explored in many prior works. The main difference appears to be the use of textual prototypes as positive and negative reference points. The authors should clarify how their approach differs from existing works and emphasize the novelty more clearly.
2. Lack of Multiple Runs: All reported experiments appear to be conducted only once. To better demonstrate the robustness of the method, the authors should run experiments using multiple random seeds and report the average results along with the standard deviation.
3. Weight Design Between Visual and Semantic Spaces: The paper lacks theoretical justification or analysis for the weighting mechanism between visual and semantic components. The authors should discuss how the weights are selected and whether any theoretical insight supports this design.
4. Typographical Error: There is a typo in the description of "chameleon" in Figure 1, where “independen-tly” is incorrectly hyphenated.
5. Ambiguity in Lines 61–71: The explanation in this section is vague. The authors should elaborate on the notions of inherent knowledge bias and inter-category semantic ambiguity. Providing concrete examples would significantly improve clarity.

---

> ### Author Rebuttal · Authors · 2025-07-31
>
> We sincerely thank the Reviewer for the thoughtful feedback and recognition of our work, particularly the novel UCIL framework and the design of the semantic prompt module. Your comments on the SAD mechanism, multiple runs, and weighting strategy are very helpful. We provide detailed responses below and will revise the paper accordingly. The full code and models will be released after publication. We hope this clarification helps the Reviewer reconsider the contributions of our work.
>
> ## Novelty of Semantic Alignment Distillation(Weakness 1)
>
> We would like to clarify that SAD is **not a simple extension of traditional prototype-based methods**, but rather a **cross-modal alignment strategy that leverages the stability of the language space to guide the evolution of the visual space**.
>
> Our proposed **hierarchical prompt template** generates high-quality and consistent class descriptions, forming a set of textual prototypes with strong discriminability and structural stability. Since the text encoder remains frozen during training, the distribution of these features remains unchanged throughout the incremental learning process. We therefore use the relative distances between textual prototypes as a set of **stable anchors**, which are projected into the visual space to constrain its drift and mitigate catastrophic forgetting.
>
> Specifically, SAD constructs a directional distillation loss (Eq. 10) by comparing each old sample’s distance to its original text prototype and to new class prototypes, thereby enforcing structural alignment between the language and visual spaces.
>
> Although SAD may appear superficially similar to certain prototype-based distillation methods in form (e.g., using class prototypes as constraints), **the fundamental design philosophy is distinctly different**. Traditional approaches typically construct prototypes in the visual space and focus on preserving or reconstructing visual feature structures. In contrast, our method **transfers semantic structure from the language space to the visual space via cross-modal alignment**. Finally, our ablation results in Table 2 (rows d → e) empirically demonstrate the significant contribution of SAD to overall performance, validating both its effectiveness and independent value.
>
> ## Lack of Multiple Runs(Weakness 2)
>
> We conducted **20 independent runs with random seeds** on the ImageNet-R dataset under three different task configurations (5 tasks, 10 tasks, and 20 tasks). We report the Top-1 average accuracy along with standard deviation as shown below:
>
> | Tasks | Mean Accuracy (%) | Std. Dev (%) |
> | :---: | :---------------: | :----------: |
> |   5   |       81.7        |     ±0.9     |
> |  10   |       82.2        |     ±0.7     |
> |  20   |       79.8        |     ±0.6     |
>
> We will include the corresponding results for CIFAR100 and CUB200 in the revised version to further validate the stability and robustness of the proposed method.
>
> ##  Theoretical Justification of the Visual-Semantic Weighting Mechanism(Weakness 3)
>
> Our design of the visual-semantic dynamic weighting mechanism is inspired by the general principle in multi-modal fusion that higher-confidence predictions tend to provide more stable and reliable supervision signals (e.g., [1]). Increasing the weight of this part will help improve the model's effectiveness. Building on this intuition, we propose a dynamic weighting strategy based on the confidence difference between modalities, enabling adaptive fusion of supervision signals from the visual and semantic spaces on a per-sample basis.
>
> As shown in Eq. (6)–(9), we compute the confidence margin between the top-1 and top-2 predicted classes within each modality to estimate its relative reliability, and assign weights accordingly to the visual and semantic pseudo labels. This mechanism allows the model to dynamically adjust the importance of multi-modal supervision based on each sample’s characteristics, thereby enhancing both discriminability and training stability. Following the setup in Section 4.1 of our paper, we conducted experiments on three incremental settings of ImageNet-R, where we replaced the confidence-based dynamic weighting strategy with a fixed weighting scheme (each weight set to 1/2). The experimental results shown in the table below demonstrate the superiority of our proposed weighting mechanism over the fixed-weight strategy.
>
> [1] Sohn, Kihyuk et al. _FixMatch: Simplifying Semi-Supervised Learning with Consistency and Confidence_, ICCV 2020.
>
> | Tasks | Dynamic weighting Acc(%) | Fixed weighting Acc(%) |
> | :---: | :----------------------: | :--------------------: |
> |   5   |           81.9           |          80.2          |
> |  10   |           82.3           |          80.3          |
> |  20   |           79.4           |          77.1          |
>
> ## Typographical Error (Weakness 4)
>
> Thank you for your careful review. We have corrected this typographical error in the revised version.
>
> ##  Clarification of term(Weakness 5)
>
> We will further clarify these two concepts and supplement them with concrete examples, as detailed below:
>
> **Inherent Knowledge Bias** refers to the tendency of large-scale pre-trained models (such as CLIP or large language models) to rely more heavily on concepts or feature patterns that appear frequently in their pre-training corpus during open-world reasoning. As a result, the models often exhibit biased predictions for low-frequency or visually indistinct categories. For instance, high-frequency animals like “cat” and “dog” are consistently recognized due to their clear visual appearance and consistent semantic expression, while less common categories such as “weasel” or “lynx” are more prone to representational drift due to their ambiguous semantics or underrepresentation in the training data. This phenomenon is closely tied to the models’ prior semantic knowledge, and partly reflects the statistical bias inherent in the “language model as knowledge base” assumption. For example, Parashar et al. (2024) [1] demonstrate that such differences in prior exposure directly lead to better recognition of frequent concepts and degraded performance on rare categories. In our initially proposed method, directly querying the LMM for each sample led to numerous incorrect labels due to this issue, which significantly degraded the model's performance.
>
> **Inter-category Semantic Ambiguity** refers to cases where different categories share highly overlapping attributes in either the visual or semantic space, making it difficult for the model to draw clear boundaries. For example, “seal” and “sea lion” are semantically distinct species, yet their visual appearances—such as body shape, texture, and typical background—are so similar that vision models often confuse them. Conversely, “octopus” and “jellyfish,” while visually very different, may be semantically conflated due to shared textual descriptors like “marine animal” or “tentacles,” bringing them closer in the text embedding space. Such within-modality ambiguities make unseen-class inference more error-prone.
>
> To address the two aforementioned issues, our method incorporates several key mechanisms:
>
> First, **to mitigate the issue of inherent knowledge bias**, we avoid directly querying the large language model (LMM) for every individual image. Instead, we design a **clustering-based representative sample selection strategy**, where only a small number (e.g.m=3) of representative images per category are selected for LMM description generation. This approach not only reduces the system’s reliance on the variability of LMM outputs, but also enhances the semantic representativeness of the generated descriptions by focusing on the visual centers of each category. Furthermore, we introduce a **hierarchical prompt template** to structurally guide the LMM in generating more stable and accurate descriptions that reflect multi-level semantic attributes. Examples of these prompts and their output consistency are provided in the appendix.
>
> Second, **to handle inter-category semantic ambiguity**, we propose a **dual-modality co-supervision framework** that leverages both visual and textual spaces for pseudo-label generation and model training. Instead of using predictions from a single modality, our method performs cross-modality alignment between the category structures in the visual and language spaces to improve discriminability. In addition, our proposed SAD mechanism leverages the stable textual prototypes from the language modality as soft constraints in the visual feature space, effectively alleviating category confusion caused by visual feature drift.
>
> In summary, our hierarchical prompt design, representative sample selection, and dual-modality co-supervision framework work in concert across generation and training phases to systematically alleviate the challenges introduced by inherent knowledge bias and semantic ambiguity.
>
> [1] Parashar, Aditya et al. _The Neglected Tails in Vision-Language Models_, CVPR 2024.

---

> > ### Comment · Reviewer_jsLU · 2025-08-05
> >
> > I appreciate the authors' efforts, and their rebuttal has addressed some of my concerns. I have raised my rating; however, I still find that there are some similarities between SAD and existing methods.

---

> ### Author Response · Authors · 2025-08-05
>
> We sincerely thank the reviewer for raising the score and recognizing the value of our work.
>
> We would like to further clarify the design of the SAD module. While it may appear superficially similar to certain prototype-based distillation approaches in terms of formulation, its essence lies in a cross-modal alignment mechanism, with a fundamentally different motivation and design philosophy.
>
> Unlike prior works that retain and replay visual features, our method is centered around leveraging the stability of the semantic (textual) space to constrain the evolution of the visual space. This semantic stability stems from two key aspects: (1) the use of prompt-based textual descriptions enhances inter-class separability in the pretrained language model, thereby improving the lower bound of semantic representation quality; and (2) the text encoder remains completely frozen throughout the incremental learning process, unaffected by streaming data. Based on this, we align the continuously updated visual features with the stable semantic structure derived from language, effectively mitigating catastrophic forgetting.
>
> We hope this helps to clearly distinguish our approach from previous works. Thank you again for your valuable feedback.

---

### Note · Authors · 2025-08-13

Dear Reviewers and Area Chair,

First, we sincerely thank all the reviewers and the Area Chair for the time and effort devoted during this review process, as well as for the careful evaluation and constructive feedback on our work.

For Unsupervised Class-Incremental Learning (UCIL), this paper fully leverages knowledge from both the visual and textual modalities, and proposes **Semantic Collaborative Facilitative Supervision** and **Semantic Alignment Distillation**. While keeping the additional computation and storage overhead extremely low, our method overcomes the challenges posed by unlabeled and dynamically evolving training data. Our method significantly surpasses the state-of-the-art (SOTA) and demonstrates strong practicality and scalability.

During the rebuttal stage, we provided systematic responses to all the questions raised by the reviewers and supplemented them with relevant experimental validation. Our explanations and experimental results were positively recognized by the reviewers, and the discussed details and suggested improvements will be further refined and presented in the final version of the paper.

Once again, we sincerely appreciate your recognition and support of our work.

---

### Decision · Program_Chairs · 2025-09-17

**Decision:**

Accept (poster)

**Comment:**

The paper tackles unsupervised class-incremental learning (UCIL), where models learn from a stream of unlabeled data without forgetting previously acquired knowledge. Traditional methods rely heavily on visual clustering, which often produces noisy pseudo-labels. This work proposes a dual-space approach that fuses visual and textual representations to generate more reliable pseudo-labels.

Strengths:
- The paper offers a compelling motivation by addressing the problem of inaccurate pseudo-labels in continual learning, clearly identifying a research gap and justifying its importance.

- The proposed method is both novel and insightful, intelligently integrating large multimodal models with a coarse-to-fine prompting strategy to deliver robust semantic supervision for unlabeled data streams.

- The method achieves strong experimental results, and ablation studies convincingly validate the importance and effectiveness of each component in the framework.

- The manuscript is exceptionally well-written and well-structured, making the content easy to follow and understand.

Weaknesses:
- The prototype-based component remains somewhat conventional compared to more radical continual learning techniques.

- It is not fully clear whether the claim that language semantics are inherently more stable than visual features is sufficiently substantiated or risks subjectivity.

Final Recommendation:

The initial reviewer ratings were mixed. During the rebuttal phase, the authors provided additional explanations and introduced new experimental results that addressed all major concerns. After rebuttal, reviewers reached a strong and consistent consensus that this paper makes significant contributions and should be accepted.